# A Deep Learning Framework for Anesthesia Depth Prediction from Drug Infusion History

**DOI:** 10.3390/s23218994

**Published:** 2023-11-06

**Authors:** Mingjin Chen, Yongkang He, Zhijing Yang

**Affiliations:** School of Information Engineering, Guangdong University of Technology, Guangzhou 510006, China; 2112103033@mail2.gdut.edu.cn (M.C.); 2112103053@mail2.gdut.edu.cn (Y.H.)

**Keywords:** depth of anesthesia, domain adaptation, knowledge distillation, target-controlled infusion

## Abstract

In the target-controlled infusion (TCI) of propofol and remifentanil intravenous anesthesia, accurate prediction of the depth of anesthesia (DOA) is very challenging. Patients with different physiological characteristics have inconsistent pharmacodynamic responses during different stages of anesthesia. For example, in TCI, older adults transition smoothly from the induction period to the maintenance period, while younger adults are more prone to anesthetic awareness, resulting in different DOA data distributions among patients. To address these problems, a deep learning framework that incorporates domain adaptation and knowledge distillation and uses propofol and remifentanil doses at historical moments to continuously predict the bispectral index (BIS) is proposed in this paper. Specifically, a modified adaptive recurrent neural network (AdaRNN) is adopted to address data distribution differences among patients. Moreover, a knowledge distillation pipeline is developed to train the prediction network by enabling it to learn intermediate feature representations of the teacher network. The experimental results show that our method exhibits better performance than existing approaches during all anesthetic phases in the TCI of propofol and remifentanil intravenous anesthesia. In particular, our method outperforms some state-of-the-art methods in terms of root mean square error and mean absolute error by 1 and 0.8, respectively, in the internal dataset as well as in the publicly available dataset.

## 1. Introduction

In recent works, clinical pharmacology research has focused on identifying best practices for ensuring patient safety by maximizing anticipated drug effects while reducing drug-induced side effects. This tradeoff is particularly important when developing anesthetic delivery patterns [1]. Specifically, the main targets are concentrated on the following three variables: analgesia (pain relief), hypnosis (based on loss of memory and consciousness), and muscle relaxation (quiescence). Therefore, it is crucial to provide appropriate doses of different drugs during this process to ensure optimal intervention conditions while preventing unnecessary and dangerous consequences.

In clinical practice, total intravenous anesthesia (TIVA) is primarily used for drug delivery. To control the depth of anesthesia (DOA), target-controlled infusion (TCI) of propofol for TIVA is one of the most widely used techniques in anesthesia and sedation. To accurately administer medication, the DOA must be continuously measured to assess the patient’s level of anesthesia and reduce drug-induced side effects. In anesthesia operations, the bispectral index (BIS) is the preferred metric for clinicians to monitor the DOA of patients. The BIS is a noninvasive system that calculates electroencephalography values ranging from 0 (no brain activity) to 100 (fully awake patients) as dimensionless numbers [2,3].

To achieve precise drug administration, various models that relate the rate of drug infusion to the DOA in humans have been proposed in recent decades. The traditional approach to target-controlled infusion is based on a pharmacokinetic and pharmacodynamic model (PK-PD) that uses propofol doses to predict propofol effect-site concentration (Ce) as an alternative to the DOA [4,5]. Although the traditional model is simple to implement, the PK-PD model fails to use the BIS as the measured metric, which is the most commonly used metric in clinical practice. More importantly, the BIS does not always agree with the model-driven Ce of propofol, especially during the anesthesia induction and recovery period [6]. With the development of machine learning, some studies [1,7,8,9,10,11] have attempted to model the DOA and the dose of propofol by using different machine learning methods. However, these models only consider the history of propofol infusion, while entirely ignoring the synergistic effect of the concomitant infusion of remifentanil on anesthesia.

In recent works, Lee et al. proposed a deep learning model for investigating the influence of the infusion histories of propofol and remifentanil and patient physiological characteristics on the DOA [12]. In addition, Sara Afshar et al. developed a combinatorial deep learning structure that predicts the depth of anesthesia according to electroencephalography (EEG) signals, demonstrating good prediction performance [13,14,15]. However, EEG signals are susceptible to inotropic and cardiac interference and sensitive to electromagnetic interference [16]. In this situation, the infusion histories of propofol and remifentanil are widely adopted to predict the DOA in clinical practice. It is well known that the statistical properties of time series change over time, thereby causing distributions to change over time. This phenomenon is known as the distribution shift problem in the field of machine learning. Our task is a time series prediction task; thus, the distribution shift problem caused by differences in the physiological characteristics of patients cannot be ignored. But, previous works fail to consider this issue. Moreover, the existing model proposed in [12] has poor performance in the induction and recovery periods, as shown in Figure 1.

To address the above two limitations, this paper proposes a new deep learning framework for DOA prediction according to drug infusion history. The key idea of the proposed approach is to use domain adaptation to address the issue that the data distribution varies across patients. Furthermore, knowledge distillation is adopted to learn the intermediate feature map of the teacher model by using the prediction model, which is obtained by the historical moment BIS as the extra feature. Specifically, an adaptive recurrent neural network (AdaRNN) [17] is applied to our task for solving the problem of data distribution differences by characterizing and matching the temporal distributions. In contrast to the traditional AdaRNN model, which uses boosting-based importance evaluations to obtain the importance vector, a neural-network-based method is used to generate the importance vector. This method can learn much useful information about the output RNN hidden states, thereby generating an enhanced importance vector. In addition, a new knowledge distillation framework is developed in our work, which uses feature-based distillation in the recurrent neural network and bottleneck networks to transfer rich temporal information and accurate efficacy information to the prediction network, respectively.

The main contributions of this work can be summarized as follows:A new deep learning framework for DOA prediction according to drug infusion history is proposed to overcome the shortcomings of the existing DOA prediction methods.A modified AdaRNN algorithm is developed in our framework for DOA prediction to address the issue of distribution shifts in the physiological characteristics of different patients.A feature-based knowledge distillation framework is proposed for time series prediction, which allows the prediction model to obtain more useful information. This framework enables the intermediate features of the model to accurately represent the DOA, thereby ensuring reliable and stable output results. To the best of our knowledge, this is the first time that knowledge distillation has been implemented to predict the DOA.Extensive experimental results show that our method has better performance than the existing DOA prediction methods on a publicly available dataset during all periods, including the induction, maintenance, and recovery periods.

The remainder of this paper is organized as follows. In Section 2, we review some related works. The proposed method is introduced in Section 3. The experimental results are discussed in Section 4. Finally, Section 5 presents the conclusions.

## 2. Related Works

### 2.1. Total Intravenous Anesthesia

The traditional TIVA method is the PK-PD model [4,5], which uses propofol doses to predict the effect-site concentration of propofol. Gonzalez-Cava et al. [18] proposed a PK-PD model that considers only propofol kinetics and is thus insufficient for accurately performing the DOA. Jose M. Gonzalez-Cava et al. [19] developed an identification algorithm based on optimization techniques and the traditional PK-PD model to determine the hypnotic level of patients under propofol–remifentanil anesthesia. In recent years, rapid advancements in machine learning and artificial intelligence have led to the development of new anesthesia prediction models. Esteban Jove et al. [7] developed a hybrid model with clustering and regression techniques that use EMG signals and propofol infusion rates to predict BIS signals. Regina Padmanabhan et al. [1] applied reinforcement learning to develop a closed-loop anesthesia controller that uses the bispectral index as the control variable while considering the mean arterial pressure (MAP). These two parameters were used to control propofol infusion rates to regulate the BIS and MAP values to the desired range. Juan A. Méndez et al. [8] proposed an adaptive model based on fuzzy logic and a genetic algorithm that aimed to provide more accurate anesthesia prediction. Sahar Javaher Haghighi et al. [9] defined a DOA index and proposed a denoising method for extracting the 40 Hz auditory steady-state response period, using adaptive multilevel wavelet denoising to calculate the proposed DOA index. It is worth noting that this algorithm does not use any medical or physiological information to define the DOA index. Therefore, this DOA index may not be well suited to patients with different physiological characteristics. Ahmad Shalbaf et al. [10] determined the best EEG features and fed these features into an adaptive neuro-fuzzy inference system with linguistic hedges to classify the patient DOA stage. However, previous work was performed in limited experimental settings using small numbers of study participants; thus, these works cannot realistically reflect the effects of drugs on the DOA or the variety of physiological characteristics of patient conditions. Lee et al. [12] proposed a deep learning model that used the infusion histories of propofol and remifentanil and patient characteristics. This deep learning model was trained on 231 subjects who received TIVA during surgery. In contrast to previous work, Sara Afshar et al. [13] combined convolutional neural networks (CNNs), long short-term memory (LSTM) networks, and an attention mechanism to develop a new framework for predicting the DOA according to EEG signals. However, in clinical surgery, anesthesiologists are more likely to control the DOA based on the effects of the drug.

### 2.2. Domain Adaptation

When the distributions of the training, validation, and test sets are markedly different, domain adaptation (DA) or domain generalization (DG) is usually used to reduce these distribution differences. Yu et al. [20] proposed a method that used a generative adversarial network (GAN) to learn transferable features and reduce distribution differences between two domains. Wang et al. [21] proposed that dynamic distribution adaptation can quantitatively estimate the relative importance of each distribution and prevent feature distortion with manifold feature learning. Li et al. [22] used maximum mean discrepancy (MMD) regularization, which reduces the distributions of different domains and matches the aligned distributions to arbitrary prior distributions through adversarial feature learning. Most DA and DG techniques apply CNNs rather than recurrent neural networks (RNNs) to classification tasks. Thus, when there are distribution differences or temporal covariate shifts in time series data, existing DA and DG methods may not be applicable. Recently, Chaehwa Yoo et al. proposed a strategy for the unsupervised DA training of sleep phase networks to overcome the domain differences between the source and target domains in more realistic situations by generating domain invariant features, and Du et al. [17] applied a novel domain adaptation approach to time series forecasting, applying temporal distribution characterization (TDC) to measure the distribution information in the time series and temporal distribution matching (TDM) to match the distributions between different periods.

### 2.3. Knowledge Distillation

Deep neural networks are adept at learning multilayer feature representations with increasing levels of abstraction. Bengio et al. [23] proposed representation learning for multilayer features. Romero et al. [24] first introduced intermediate representations in Fitnets, which aimed to provide the output of intermediate teacher networks to improve the training of the student model. Inspired by this approach, Gao et al. [25] distilled the feature maps of different levels of the teacher and student networks. Hong et al. [26] trained teacher networks using heterogeneous tasks and applied knowledge distillation to the intermediate layer representations to ensure consistency between the intermediate feature representations of the reconstruction processes of the student and teacher networks. Zhang et al. [27] distilled features in each level of the teacher and student networks, thereby allowing the student network to learn privileged information from the teacher network. Zhang et al.’s model [27] was applied to time series prediction, demonstrating the effectiveness of knowledge distillation in time series prediction tasks.

## 3. Methodology

Different from the previous DOA prediction technique, our proposed method uses domain adaptation and knowledge distillation techniques to further improve the DOA prediction accuracy. Specifically, for domain adaptation, a modified AdaRNN is utilized to enhance the generalizability of the proposed model. For the knowledge distillation, the teacher network aims to provide intermediate feature representations of the BIS, while the student network aims to predict the BIS more precisely by transforming the intermediate features in the teacher’s intermediate feature domain. Figure 2 shows an overview of our proposed framework, which consists of a teacher model and a student model. The teacher model has an additional input to learn an excellent representation of the BIS and transfer the BIS representation via knowledge distillation. Moreover, the distribution shift problem caused by the various physiological characteristics of different patients is also solved by modifying AdaRNN to improve the robustness and generalizability of the proposed DOA prediction model.

### 3.1. AdaRNN

AdaRNN is mainly adopted to solve the temporal covariate shift (TCS) problem. AdaRNN can characterize the temporal distributions and capture the long-term time dependencies in time series. In general, AdaRNN includes temporal distribution characterization (TDC) and temporal distribution matching (TDM). In order to take advantage of AdaRNN for our task, the TDM approach in AdaRNN is modified to efficiently adapt the proposed network. Figure 3 illustrates an overview of the TDC approach, which divides the data into regions with the largest distribution differences, thereby ensuring that the model is trained starting from the worst case. Figure 4 depicts an overview of the TDM method, which reduces cross-domain distribution differences in the model to enhance the generalizability of the model.

#### 3.1.1. Temporal Distribution Characterization

According to the maximum entropy (ME) principle [28], it is reasonable to have as diverse a distribution as possible for each period to maximize the entropy of the total distribution, without any prior assumptions about how to divide the time series data. This approach allows for more general and flexible modeling of future data. In addition, the use of shared knowledge in the time series data can be maximized by identifying periods that are least similar to each other. Thus, the problem can be expressed as follows:(1)$$\begin{array}{c} \max_{0<K\leq K_{0}}\max_{n_{1},\cdots ,n_{k}}\frac{1}{K}\sum_{1\leq i\neq j\leq K}d\left(D_{i},D_{j}\right)\\ \;s.t.\;\forall i,\Delta_{1}<\left|D_{i}\right|<\Delta_{2};\sum_{i}D_{i}=n \end{array}$$
where d(·,·) is a distance metric, $\Delta_{1}$ and $\Delta_{2}$ are predefined parameters, $K_{0}$ is a hyperparameter, and $D_{i}$ denotes the *i*-th time series segment.

The goal of the optimization problem (1) is to maximize the difference in the distribution in each period by determining *K*. If the model learns from the worst-case scenario, it has better generalizability to the unseen test data. This assumption has been verified in theoretical analyses of the time series models [29,30], showing that diversity is crucial in time series modeling. The time series partitioning of the optimization problem in (1) is a computationally intractable problem that may not have a closed solution. However, the optimization problem in (1) can be solved by using dynamic programming (DP) [31] with a suitable distance metric. Considering the issues of scalability and efficiency for the large-scale data, similar to previous methods, a greedy algorithm is adopted for the AdaRNN in our work to solve the optimization problem in (1). To show the flow of TDC more clearly, the pseudo-code is summarized in Algorithm 1.
**Algorithm 1:** Temporal distribution characterization
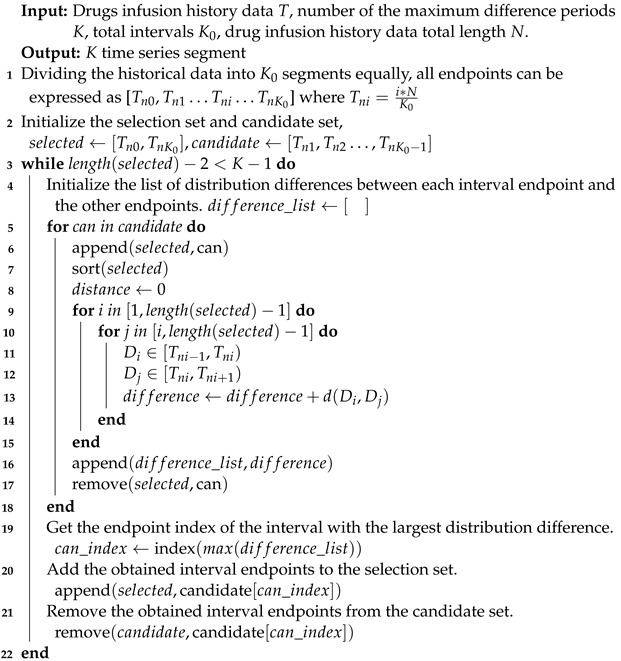


#### 3.1.2. Temporal Distribution Matching

Once the TDC approach determines the time periods to be learned, the temporal distribution matching approach is required, which learns the public knowledge shared across each time period by matching the distributions of different time periods. Thus, the learned model *M* has better generalizability on the test data than other methods that only rely on the local or statistical information.

The TDM prediction loss $L_{pred}$ can be calculated as follows:(2)$$L_{\mathrm{pred}}\left(\theta \right)=\frac{1}{K}\sum_{j=1}^{K}\frac{1}{\left|D_{j}\right|}\sum_{i=1}^{\left|D_{j}\right|}l\left(y_{i}^{j},M_{t}\left(x_{i}^{j};\theta \right)\right)$$
where $\left(x_{i}^{j},y_{i}^{j}\right)$ denotes the i-th labeled segment in period $D_{i}$, l(,) is a loss function, e.g., the mean square error (MSE) loss, θ denotes the learnable model parameters, and $M_{t}$ denotes the teacher model.

In contrast, the minimization of Formula (2) can learn only the predictive knowledge for each period and cannot reduce distribution differences between each period to exploit common knowledge. A primitive approach is to use some distribution measure of distance as a regularization term for each period pair $D_{i}$ and $D_{j}$. Based on previous domain adaptation research [32,33] in which distribution matching is typically performed on high-level representations, Du et al. [17] applied a distribution matching regularization term to the final output of the RNN model.

Since each hidden state only contains the part of the distribution information of an input sequence, Du et al. [17] adopted the distribution regularization term in each hidden state of the RNN model to capture the temporal dependency of each hidden state. Then, they introduced the importance vector α, which aims to learn the relative importance of each hidden state, with each hidden state weighted with a normalized α.

Given a period pair $\left(D_{i},D_{j}\right)$, the temporal distribution matching loss can be formulated as:(3)$$L_{tdm}\left(D_{i},D_{j};\theta \right)=\sum_{t=1}^{V}\alpha_{i,j}^{t}d\left(h_{i}^{t},h_{j}^{t};\theta \right)$$
where $\alpha_{i,j}^{t}$ denotes the distribution importance between periods $D_{i}$ and $D_{j}$ in state *t*, and *V* denotes the *V* hidden states of the RNN.

By integrating (2) and (3), the final temporal distribution matching loss can be formulated as
(4)$$\begin{array}{c} L\left(\theta ,\alpha \right)=L_{pred}+\lambda \frac{2}{K(K-1)}\sum_{i,j}^{i\neq j}L_{tdm}\left(D_{i},D_{j};\theta ,\alpha \right) \end{array}$$
where λ is a balance hyperparameter.

#### 3.1.3. Neural-Network-Based Importance Evaluation

Du et al. [17] used a boosting-based importance evaluation algorithm to learn α. However, in our experiments, we observe that the boosting-based method is difficult to improve the experimental results. In contrast, a naive method that obtains α through a fully connected network usually results in good results. The boosting-based method is not suitable for our task since α and θ are highly relevant in the early training phase and the hidden state representations are learned according to the model parameter θ. Thus, α must be determined according to θ. Therefore, a neural network is used in our proposed framework to comprehensively explore the importance of each hidden state. Specifically, the importance of the RNN output hidden states is initialized as the mean value of the RNN output hidden states. The initialization can be expressed as $\alpha_{i,j}={\left\{1/V\right\}}^{V}$. Then, we use a fully connected network with weight $W_{i,j}$ that takes $\left(H_{i},H_{j}\right)$ as input and outputs α. This process can be formulated as $\alpha_{i,j}=g\left(W_{i,j}⊙{\left[H_{i},H_{j}\right]}^{T}\right)$, where *g* denotes an activation and normalization function and ⊙ stands for the elementwise product.

### 3.2. Knowledge Distillation

Since response-based knowledge is mainly used in classification tasks and our task is a regression task, we use a knowledge distillation framework based on feature representations to enable the student network to learn better intermediate feature representations from the more informative teacher network. Previous work on feature-based knowledge distillation [25,26,27] has shown that teacher knowledge is typically propagated through high and low levels. Thus, in our method, the knowledge is applied to propagate through the final outputs of the RNN and bottleneck network to learn the temporal relationships in the teacher RNN and the feature map representations of the teacher bottleneck network, respectively.

Firstly, we introduce the historical moment BIS data (i.e., BIS data before the observation moment) in the teacher network to learn the DOA representation. The student network uses the propofol and remifentanil infusion histories as the inputs. Secondly, to propagate the knowledge from the teacher network to the student network more effectively, the hidden states of the output of the student RNN are used to mimic the hidden states of the output of the teacher RNN in order to improve the prediction performance of the student network. Moreover, the feature map of the bottleneck of the student network is adopted in our framework to mimic the feature map of the bottleneck of the teacher network. In this situation, the similarity between the predictions of the student and teacher networks will be increased.

#### 3.2.1. Knowledge Distillation via RNN

Since the teacher network introduces the historical moment BIS data, the output hidden states of the teacher RNN contain more richer temporal information about the DOA, thus allowing the model to predict the BIS change trends more accurately. Therefore, the output hidden states of the student RNN is utilized in our proposed framework to imitate the output hidden states of the teacher RNN. In contrast to the TDM regularization term, we imitate the feature representations of the output hidden states of the teacher RNN to learn the temporal relationships of the teacher RNN for improving the prediction performance of the student network. Therefore, the feature imitation loss is introduced. Specifically, we use a gated recurrent unit (GRU) as the basic unit of the RNN module. Assume that $T_{gru}\left(z\right)$ is the feature representation of the teacher GRU trained on *z*, which denotes the input data to the teacher network and includes historical BIS data, and $S_{gru}\left(x\right)$ denotes the feature representation of the student GRU trained on *x*, which includes the propofol and remifentanil infusion histories. $L_{g}\left(\theta \right)$ can be formulated as follows:(5)$$L_{g}\left(\theta \right)=\left||T_{gru}\left(z;\theta \right)-S_{gru}\left(x;\theta \right)\right|{|}_{2}$$

#### 3.2.2. Knowledge Distillation via Bottleneck

For the RNN output feature, only the effects of the propofol and remifentanil on the BIS are considered. However, according to the experimental results in [34], the drug elimination rates of the remifentanil and propofol were different among people with different physiological characteristics (e.g., older people with the same duration of surgery had higher drug elimination rates). In contrast to the temporal information contained in RNNs, the bottlenecks add the static physiological information (e.g., height, weight, sex, and age) into the RNN output to fully learn the anesthetic effects of both propofol and remifentanil drugs on people with different physiological characteristics. Moreover, the teacher network uses the historical BIS data features, which can accurately represent the effects of static physiological information and dynamic drug information on the BIS. Thus, the feature representations of the student bottleneck output is utilized to mimic the feature representations of the teacher bottleneck output. Specifically, supposing that $T_{b}\left(z\right)$ denotes the feature representations of the teacher bottleneck trained on *z* and $S_{b}\left(x\right)$ denotes the feature representations of the student bottleneck trained on *x*, $L_{b}\left(\theta \right)$ can be formulated as follows:(6)$$L_{b}\left(\theta \right)=\left||T_{b}\left(z;\theta \right)-S_{b}\left(x;\theta \right)\right|{|}_{2}$$

### 3.3. Loss Function

#### 3.3.1. Loss Function of the Teacher Network

To obtain more effective feature representations for the teacher network, combining (2) and (3), the teacher loss can be formulated as follows:(7)$$\begin{array}{c} L_{T}\left(\theta ,\alpha \right)=L_{pred}+\lambda \frac{2}{K(K-1)}\sum_{i,j}^{i\neq j}L_{tdm}\left(D_{i},D_{j};\theta ,\alpha \right) \end{array}$$

#### 3.3.2. Loss Function of the Student/Prediction Networks

The loss of the prediction network has three components: the predicted loss, the TDM regulation loss, and the distillation loss (i.e., $L_{g}\left(\theta \right)$ and $L_{b}\left(\theta \right)$), which can be formulated as follows:(8)$$\begin{array}{c} L_{S}\left(\theta ,\alpha \right)=L_{\mathrm{pred}\;}+\lambda \frac{2}{K(K-1)}\sum_{i,j}^{i\neq j}L_{tdm}\left(D_{i},D_{j};\theta ,\alpha \right)\\ \;\;\;+L_{g}\left(\theta \right)+L_{b}\left(\theta \right) \end{array}$$

In our method, the predicted loss is used to learn the DOA during each period, and the TDM loss is adopted to reduce the distribution differences between various periods. Then, the effective DOA feature representation in the teacher model is transferred to the student model by the distillation loss.

## 4. Experiments

### 4.1. DOA Type and Dataset

In order to calculate the DOA, some typical measurements have been proposed in clinical anesthesia, which are shown in Table 1. Compared with other measurements of DOA type (e.g., Narcotrend index and Patient State index), the BIS has two main advantages. On the one hand, the BIS is the most widely used anesthesia depth indicator and has been approved by the FDA to be marketed as a monitor of anesthetic effects on the brain. The BIS correlates well with the sedative effects of many anesthetic drugs and can accurately reflect their depth of sedation, especially for commonly used anesthetic drugs such as propofol and sevoflurane. On the other hand, the BIS has been widely adopted in clinical validation. Particularly, the BIS is collected as the only indicator of anesthesia depth in the publicly available dataset (i.e., VitalDB). Therefore, the BIS is adopted as the measurement of DOA type in our work.

The dataset used in the experiments is the VitalDB (https://vitaldb.net) (accessed on 1 January 2022). database, which is collected and registered by the Department of Anesthesiology and Pain Medicine, Seoul National University Hospital, Seoul Metropolitan Medical College, Seoul, Republic of Korea [40]. The VitalDB database is an open-access dataset, which can freely download from the website, https://vitaldb.net (accessed on 1 January 2022). In addition, the VitalDB database is a comprehensive dataset that includes the intraoperative biosignals and clinical information of 6388 surgical patients. It not only contains the demographic data (height, weight, sex, and age), but also contains more than 60 procedure-related clinical indicators of basic equipment used in operating rooms, such as patient monitors, anesthesia machines, and BIS monitors, as well as target-controlled infusion pumps, cardiac output monitors, and local oximeters. The BIS data include the BIS values and the signal quality indices collected by BIS VISTA at 1 s intervals. The propofol and remifentanil data include the cumulative infusion volumes, Ce values, and plasma concentration (Cp) values of the two drugs collected by target-controlled infusion pumps at 1 s intervals.

In our experiments, the data of 1000 patients are randomly selected from the VitalDB. After data processing, only 332 data are retained and the rest are discarded due to the excessive missing BIS or the drug administration records. The detailed data processing is given in Section 4.2. The retained 332 patients are randomly divided into training, validation, and testing datasets, in which the training, validation, and testing dataset contain 180, 76, and 76 cases, respectively. The characteristics of these three datasets are shown in Table 2.

### 4.2. Data Processing

From the start of the propofol or remifentanil infusion to the end of the BIS measurement, the data meeting the following conditions are discarded:If the BIS value at the start of the drug infusion is less than 80;If the data are missing for more than 300 s;If the first BIS is recorded when the cumulative amount of the infused drug is not 0.

After the above processing steps, the patient sample data may still contain a small number of missing or erroneous values. To address the issue of missing values in data, according to the linear relationship between propofol and remifentanil over time, the missing values in the patient data are determined by linear interpolation. Since the total dosage of propofol and remifentanil is monotonically nondecreasing, for cases in which there is a partial decrease in the cumulative infusion volumes in the patient data, the erroneous values may be caused by instrument recordings or how the data are exported. To solve this problem, the mean values of the moments before and after the erroneous moment are used. The above processing techniques were applied to the training set.

The cumulative injection volume is recorded by target-controlled infusion pumps, and the injection history data collected by the target-controlled infusion pumps are updated every 10 s [41]. Therefore, the cumulative medication usage of propofol and remifentanil over 10 s is adopted as a feature to create a time window with a sequence length of 120. Since the data variations among different patients are useless for training the network, we downsample the training data, i.e., one sampling point every 10 s is selected. Since the BIS values of different patients have diverse trends and the BIS values of each patient change frequently, we smoothed the BIS values in the training set by using locally weighted scatter plot smoothing (LOWESS) with a smoothing parameter of 0.03 to reduce computational errors during training. The unprocessed BIS values are used in the validation and test datasets.

### 4.3. Experiment Settings

#### 4.3.1. Teacher and Student Network

The teacher and student networks have the same structure, except that the input to the teacher network has an additional feature (the BIS value of the previous moment) that is not available to the student network. Our network includes an RNN module, a bottleneck module, and an output module. We use a GRU as the basic unit in the RNN module because of its high performance in practical applications. The bottleneck module focuses on learning the anesthetic effects of propofol and remifentanil on patients with different physiological characteristics. In our work, the bottleneck module consists of two fully connected networks. After the bottleneck network, our proposed model learns the combined dynamic (propofol and remifentanil drug dosage information) and static (patient physiological characteristics) information. Finally, the predicted BIS values with a fully connected network are obtained.

#### 4.3.2. Implementation Details

In the experiments, PyTorch 1.7.1 is used to implement our proposed model with a 24 GB NVIDIA TITAN RTX GPU and an Intel(R) Xeon(R) CPU E5-2680 v3 @ 2.50 GHz with 32 GB RAM and 128 GB SSD, and our teacher and student networks are trained simultaneously. The network is trained by an Adam optimizer with an initial learning rate of 0.005. The learning rate is decayed every 10 epochs by a factor of 0.1, and the weight clipping is utilized to prevent gradient explosion. We use a batch size of 256 in the training phase and a batch size of 128 in the validation and testing phases. The balance parameter λ is set to 0.05 and the number of difference distribution periods K is set to 5. The cosine similarity is adopted to measure the distribution differences across each period. The entire training process takes no more than 2 h, where the training batch size is set to 256. Our code is available at https://github.com/chanwendy/Domain-adaptation-DOA-prediction (accessed on 25 June 2023).

### 4.4. Evaluation Metrics and Results

The experimental results are expressed as the regression errors. To evaluate the regression performance of our proposed framework, the predicted values are assessed according to the following commonly used evaluation metrics [42,43]: the mean absolute error (MAE) [44] and the root mean square error (RMSE) [44]. Based on [28], the performance error (PE), median performance error (MDPE) [28], and median absolute performance error (MDAPE) [28] are also used to measure the performance for comparison. Particularly, the PE is calculated as the measured BIS-predicted BIS/predicted BIS. Since the PE is based on performance error as part of the predicted drug concentration, it is particularly useful to clinicians. The MDPE is a signed value and therefore indicates the direction of the performance error (overprediction or underprediction) rather than the size of the error. The MDAPE measures the computer-controlled infusion pump (CCIP) performance and reflects the inaccuracy of the CCIP. To further reveal the performance of the model, we evaluated the performance of the three periods (induction, maintenance, recovery) in TIVA. The induction period in TIVA represents the 10 min period starting with the surgical injection of propofol, and the recovery period represents the period between stopping the surgical injection of propofol to the end of anesthesia; the remaining time represents the maintenance period.

#### 4.4.1. Open-Access Dataset

In the experiment, we conduct a comprehensive analysis of mean comparisons across all periods, including the induction period, the maintenance period, and the recovery period. The compared methods are, respectively, the baseline method [12], FEDformer [45], and Crossformer [46]. The experiment results are shown in Table 3, and are depicted as the mean ± standard deviation. According to Table 3, one can observe that, in most situations, our proposed method has better performance than other compared methods during each period of anesthesia. This indicates that our proposed method can efficiently solve these issues of the baseline method, i.e., the weak generalizability of the model and the poor performance in the induction and recovery periods. In addition, we use the vanilla transformer [47] as the transformer backbone of our pipeline; the results are shown in Table 3. Benefiting from the long-term modeling capability of the transformer model, it can effectively capture the global features of the time series data, which can effectively learn the continuous effect of anesthesia drugs on the human body; therefore, the transformer encoder can obtain more accurate features of the depth of anesthesia. In addition, there is only one case in which FEDformer outperforms the proposed model in the recovery period. The main reason may be that FEDformer is a transformer-based method that can better capture the global features of the time series and thus is more able to synthesize information before the recovery period. However, since FEDformer fails to take into account the excessive differences in the distribution of each individual, the experimental results in the final evaluation metrics (e.g., RMSE and MAE) of anesthesia prediction are worse than those of ours.

Figure 5 shows the results of the visual comparison between the proposed method and the compared methods. From this figure, it can be seen that the proposed method can predict the DOA of unknown patients more accurately than these compared methods. The main reason that our model outperforms the compared models is that the proposed framework considers the distribution shift problem caused by different patient data and solves this problem with an appropriate method. Moreover, the introduction of historical BIS values to the teacher network in the distillation model allows the prediction model to obtain much useful information about the intermediate feature representations of the BIS values, thereby allowing the model to predict partial mutations in the induction and recovery periods.

#### 4.4.2. In-House Dataset

To further validate the generalization ability of our model and the applicability of our model under different measurements of DOA type, we perform the training and testing on our in-house dataset. This dataset uses the Narcotrend index (NI) as the measurement for the depth of anesthesia. Compared with the BIS, the NI has a larger response gap for patients of different age groups and different disease states, creating a larger domain shift. The main difference between the VitalDB dataset and our in-house dataset is shown in Table 4.

In the experiment, we randomly select 24 cases as the training set, and others as the test cases. The experiment results are shown in Table 5. Obviously, our proposed method still performs better than these compared methods during each period of anesthesia. This demonstrates that our proposed method can achieve good results when there is a large domain shift in the dataset, while other compared methods suffer from significant performance degradation due to the domain shift. In addition, the results without the preprocessed data of our in-house dataset show that noisy data will lead to performance degradation, but our method can still achieve better performance than the SOTA models using the preprocessed data for training.

Figure 6 shows the results of the visual comparison between the proposed method and these compared methods in our in-house dataset. From this figure, it can be seen that, compared with other compared methods, our proposed method has better trend and prediction results in all periods. This is because our proposed method is not only trained from the interval with the largest distribution differences but also utilizes the NI data of historical moments to the teacher network.

#### 4.4.3. Statics Test and Application in the Real Word

In addition, we perform some statistical tests, such as pair-*t* test and F-test, with these compared methods in both public and in-house datasets, and the experimental results are shown in Table 6. According to this table, one can see that, in most situations, the *p*-values for the pair-*t* test and F-test are less than 0.05 in both datasets, which indicates that our proposed model has better performance than other compared methods from the viewpoint of statistical test. There is only one case where the *p*-values for the Crossformer in the F-test are greater than 0.05. The main reason may be that the Crossformer takes into account the information interaction between multiple variables and achieves relatively good stability. However, from the results in Table 3, one can see that, in most cases, the standard deviations of our proposed model are still better than Crossformer.

In a real clinical environment, the closed-loop target-controlled infusion system is divided into two main parts, which are the prediction of DOA and the control of drug dosage according to the DOA, respectively. Our work mainly addresses the first part of predicting the DOA based on drug efficacy, and the part of controlling the dose based on the DOA is our future work. The prediction of DOA is commonly performed at a frequency of one prediction per second, aiming to estimate the BIS value. To simulate this inference process and evaluate the real-time capabilities of our model, we set the batch size to one in the experiments. Remarkably, our findings reveal that our model achieves a remarkable inference time of only 1 × 10^−5^ s per prediction of the BIS value. In addition, we compute model complexity between our model and the compared methods; the results are shown in Table 7. One can see that our model has a smaller parameter and FLOPS compared with the SOTA models. Although the baseline model parameter and FLOPS are the smallest, its ability is also the worst. In contrast, although our model increases the number of parameters with respect to the baseline, at the same time, it obtains a better performance improvement that outperforms the SOTA models, and our proposed model predicts the depth of anesthesia in only 1 × 10^−5^ per second, which can fully meet the clinical requirements. Therefore, our proposed model can be suited for real-time monitoring applications.

### 4.5. Ablation Study

To verify the effectiveness of the neural-network-based importance evaluation method, a comparative experiment with our TDM method and the boosting-based TDM method is derived to evaluate the respective performance. Table 8 shows the experimental results of our TDM method and the boosting-based method. Compared with the baseline method in Table 3, our TDM method and the boosting-based TDM method both have better experimental results than the baseline. From Table 8, one can see that our TDM method outperforms the boosting-based method in all periods, including the induction and maintenance periods. In addition, the performance between our TDM method and the boosting-based TDM method is slightly different in the recovery period. This is because the boosting-based method focuses more on the distribution at the end of the time series, so it has better results in the recovery phase, while our neural-based method focuses more on the early part of the time series, i.e., the induction period and maintenance period. This only causes our method to be a little bit worse than the boosting-based method in the recovery period. However, our method achieves significantly better results in the induction period, the maintenance period, and all three periods combined. Therefore, the proposed TDM method is adopted in our task.

The experimental results of the ablation study are illustrated in Figure 7. As can be seen from Figure 7, when the baseline is introduced with the AdaRNN, the proposed model usually works better than the original model in all stages of anesthesia. This is due to the fact that the proposed model trains in the worst case, which can alleviate the effects of excessive differences in the distribution of different patient data to a certain extent, making the model more robust and generalized. Thus, the model is able to predict better in the face of unknown patient data.

According to Figure 7a, when a knowledge distillation approach is applied in the baseline model, the new model has better performance during each period. In Figure 7c,d, the MDPE achieves better performance in the induction and maintenance periods than the AdaRNN model, which indicates that the predicted values for these two periods are more accurate and display fewer fluctuations. Therefore, knowledge distillation is useful for solving the overprediction problem faced by the AdaRNN model in the induction and maintenance periods. Based on Figure 7a,b, although incorporating the knowledge distillation into the baseline model has good stability, the difference between the AdaRNN and the knowledge distillation approach is very small in terms of the RMSE and MAE. However, when the AdaRNN and knowledge distillation are used in the same model, the prediction performance can improve, and the combined model still brings good stability. Thus, incorporating knowledge distillation into the AdaRNN model results in a great improvement in our proposed work.

## 5. Conclusions

Accurate anesthesia depth prediction is crucial in clinical surgeries. In this paper, a new deep learning framework is proposed that combines the domain adaptation technique and the knowledge distillation technique to improve the generalizability and prediction accuracy of anesthesia depth prediction models. Specifically, different from previous works, the proposed framework first considers the problem of distribution shifts among patients with different physiological characteristics data and trains the model from cases with the largest distribution differences in order to enhance the generalizability of the proposed model. Moreover, to improve the accuracy of anesthesia depth prediction, the BIS values of the historical moments are introduced in the teacher network of the proposed method to extract the intermediate features, representing the anesthesia depth accurately. After that, the prediction network is utilized to mimic the feature representations of the teacher network. Experimental results have shown the effectiveness of our proposed framework on a publicly available dataset compared with the traditional response model and the baseline model.

In future studies, more information on historical drug doses or patient physiological indicators related to clinical anesthesia, such as plasma concentration, effect-site concentration, target concentration, and infusion rate, can be added to our model to further improve the prediction accuracy. Furthermore, an attention mechanism can be incorporated to allow the model to focus on anesthesia depth features that are effective for predicting anesthesia depth. Finally, due to the imbalance in the quantity of data (i.e., most of the BIS is in the range of 35–45, while the other quantities are sparse). Thus, addressing data imbalance in our task also can improve the performance.

## Figures and Tables

**Figure 1 sensors-23-08994-f001:**
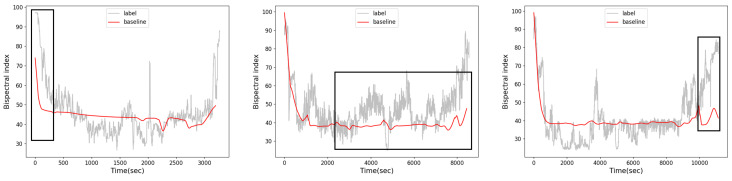
Visualization of the results of the baseline method [12]. The baseline method performs poorly during the induction and recovery periods. In addition, the baseline model predicts that the patient’s BIS is approximately 40 during the maintenance period. However, some patients have BIS values of approximately 50 or 30 during the maintenance period.

**Figure 2 sensors-23-08994-f002:**
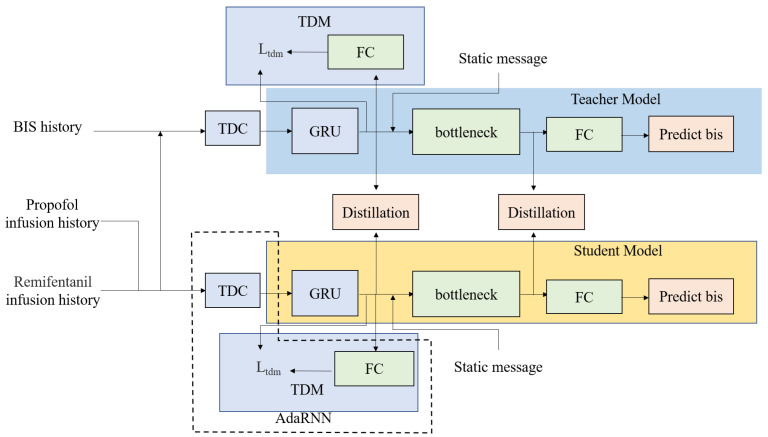
An overview of our deep learning framework for DOA prediction. The teacher model has an extra input (i.e., the BIS history), which allows the teacher model to learn a more accurate representation of the DOA and transfer the DOA representation to the student model through various kinds of layers by knowledge distillation. The AdaRNN model includes temporal distribution characterization (TDC) and temporal distribution matching (TDM), which are used to determine the K-segment intervals with the largest distribution differences and reduce cross-domain distribution differences according to the drug infusion history, thereby improving the generalizability of the model. GRU denotes the gated recurrent unit, and FC denotes the fully connected network.

**Figure 3 sensors-23-08994-f003:**
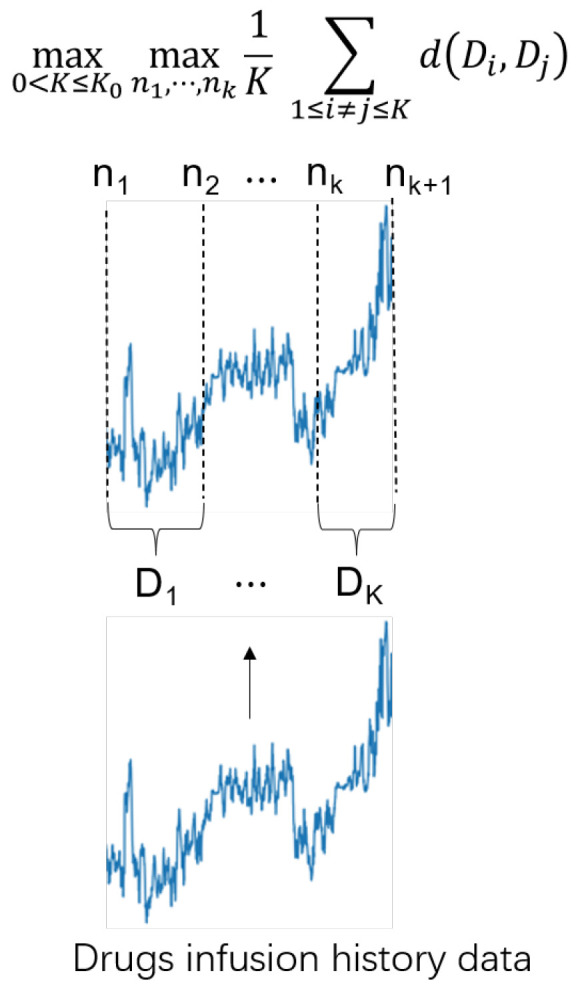
An overview of the temporal distribution characterization (TDC) approach, which divides the drug infusion history data into K intervals and obtains the largest distribution difference between every two intervals.

**Figure 4 sensors-23-08994-f004:**
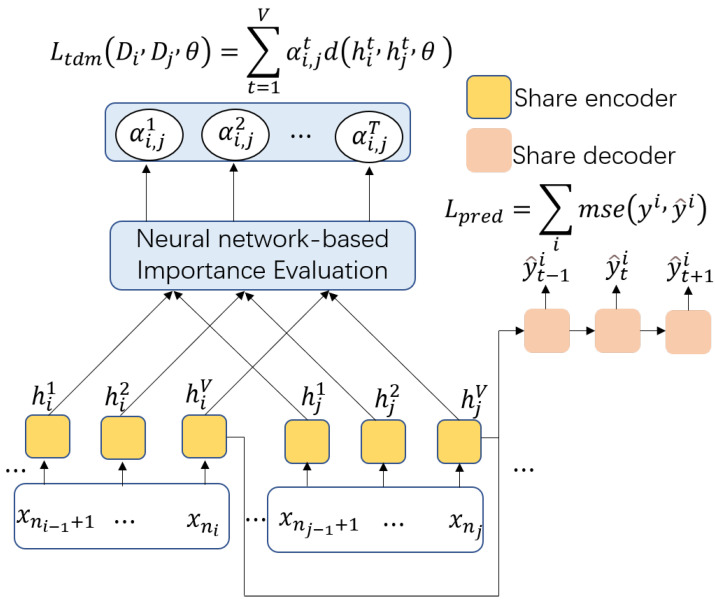
Overview of the temporal distribution matching, which reduces cross-domain shifts in the K intervals according to the GRU output.

**Figure 5 sensors-23-08994-f005:**
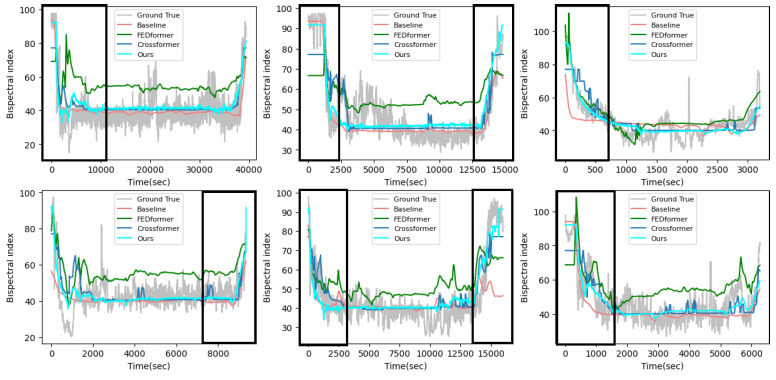
Visualization of the test cases with other compared methods in the VitalDB dataset.

**Figure 6 sensors-23-08994-f006:**
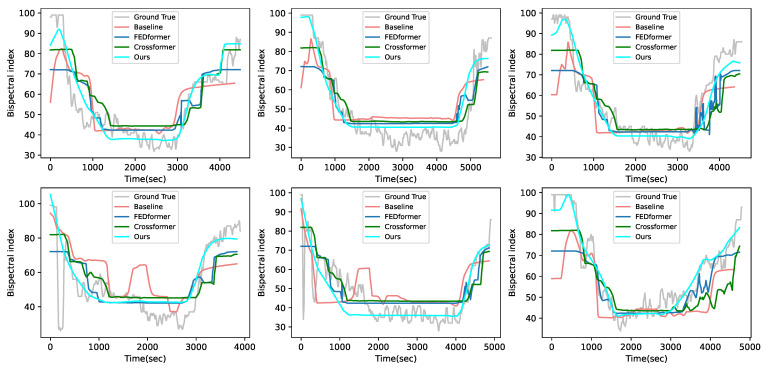
Visualization of the test cases with other compared methods in our in-house dataset.

**Figure 7 sensors-23-08994-f007:**
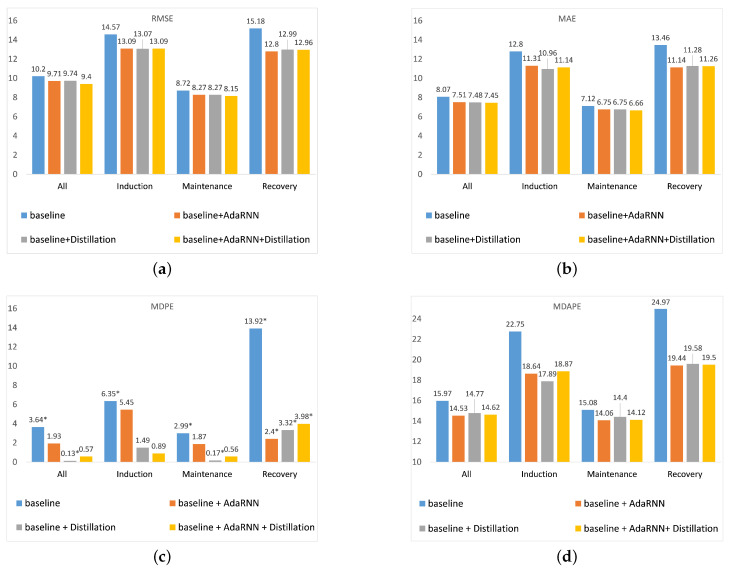
Ablation analysis of the various methods in all periods for different evaluation metrics. (**a**) RMSE; (**b**) MAE; (**c**) MDPE, where * denotes a negative value; (**d**) MDAPE.

**Table 1 sensors-23-08994-t001:** Different measurements of DOA type.

DOA Type	
Bispectral Index [35]	Narcotrend Index [36]
Phase Lag Entropy [37]	Entropy [35]
SedLine [38]	Patient State Index [39]
Auditory Evoked Potential [35]	Surgical Stress Index [35]

**Table 2 sensors-23-08994-t002:** Patient characteristics, mean ± standard deviation (min-max).

	Training Data Set	Validation Data Set	Testing Data Set
N	180	76	76
Age (yr)	56.1 ± 14.0 (17–82)	56.3 ± 15.0 (17–79)	56.2 ± 15.1 (17–79)
Sex (male/female)	113/67	47/29	40/36
Weight (kg)	61.5 ± 10.2 (37.9–98.1)	60.7 ± 10.3 (37.9–98.1)	60.0 ± 9.8 (37.9–81.6)
Height (cm)	163.2 ± 8.2 (138.8–186.6)	162.3 ± 7.9 (138.8–182.0)	161.2 ± 7.5 (138.8–182.0)
Median BIS	41.1 ± 5.4 (25.9–59.5)	43.1 ± 6.1 (23.1–57.2)	42.5 ± 5.8 (30.6–55.9)
Propofol total dose (g)	1.19 ± 0.63 (0.28–3.41)	1.27 ± 0.71 (0.32–3.31)	1.32 ± 0.71 (0.30–4.24)
Propofol median Ce (μg/mL)	3.02 ± 0.47 (1.91–4.30)	3.06 ± 0.49 (2.00–4.00)	3.05 ± 0.50 (1.60–4.00)
Remifentanil total dose (g)	1.46 ± 1.01 (0.29–6.29)	1.43 ± 0.84 (0.25–3.70)	1.46 ± 0.91 (0.34–5.16)
Remifentanil median Ce (μg/mL)	3.73 ± 1.08 (1.50–6.01)	3.67 ± 0.95 (2.00–6.97)	3.70 ± 0.87 (2.00–6.00)

**Table 3 sensors-23-08994-t003:** Comparison of the metrics between our model and the compared model during three anesthesia periods in vitalDB dataset with BIS as the DOA.

	RMSE					MAE				
Anesthesia Period	Baseline [12]	FEDformer [45]	Crossformer [46]	Ours	Ours (Transformer)	Baseline [12]	FEDformer [45]	Crossformer [46]	Ours	Ours (Transformer)
All	10.20 ± 2.45	10.08 ± 2.40	9.66 ± 2.37	9.40 ± 2.29	**9.07** ± 1.76	8.07 ± 2.56	7.75 ± 2.39	7.61 ± 2.34	7.45 ± 2.30	**7.29** ± 1.40
Induction	14.57 ± 3.39	13.75 ± 5.00	13.64 ± 4.66	13.09 ± 3.90	**10.91** ± 3.93	12.80 ± 3.79	11.45 ± 4.63	11.63 ± 4.39	11.14 ± 3.69	**9.00** ± 3.54
Maintenance	8.72 ± 2.78	8.56 ± 2.55	8.37 ± 2.62	**8.16** ± 2.51	8.63 ± 2.06	7.12 ± 2.79	6.95 ± 2.58	6.80 ± 2.61	**6.66** ± 2.52	7.00 ± 1.60
Recovery	15.18 ± 6.43	13.30 ± 5.43	13.16 ± 5.90	12.96 ± 5.45	**10.46** ± 2.94	13.43 ± 6.43	11.30 ± 5.13	11.43 ± 5.67	11.26 ± 5.22	**8.74** ± 2.45
	**MDPE (%)**					**MDAPE (%)**				
Anesthesia Period	Baseline [12]	FEDformer [45]	Crossformer [46]	Ours	Ours (Transformer)	Baseline [12]	FEDformer [45]	Crossformer [46]	Ours	Ours (Transformer)
All	−3.64 ± 14.96	1.76 ± 13.84	−0.59 ± 14.05	**0.57** ± 13.77	−13.89 ± 4.98	15.97 ± 7.91	15.35 ± 7.61	14.83 ± 6.66	**14.62** ± 6.40	16.76 ± 3.53
Induction	−6.35 ± 20.50	7.72 ± 16.84	6.79 ± 17.73	**0.89** ± 18.54	−3.15 ± 14.30	22.75 ± 9.39	21.90 ± 19.50	18.88 ± 8.69	18.87 ± 8.14	**15.64** ± 9.40
Maintenance	−2.99 ± 15.24	1.17 ± 14.31	−0.97 ± 14.48	**0.56** ± 14.10	−15.20 ± 5.51	15.08 ± 8.34	14.96 ± 8.01	14.44 ± 8.37	**14.12** ± 8.97	17.27 ± 4.14
Recovery	−13.92 ± 25.44	**1.94** ± 19.35	−2.33 ± 21.69	−3.98 ± 21.38	−7.44 ± 11.56	24.97 ± 16.03	19.75 ± 12.90	19.81 ± 11.10	19.52 ± 11.18	**15.34** ± 6.22

**Table 4 sensors-23-08994-t004:** Difference in VitalDB and our dataset.

	Our Dataset	VitalDB (Training Set)
N	44	180
Age (yr)	39.9 ± 13.4 (19–69)	56.1 ± 14.0 (17–82)
Sex (male/female)	22/22	113/67
Weight (kg)	62.6 ± 10.5 (43–105)	61.5 ± 10.2 (37.9–98.1)
Height (cm)	166 ± 8.1 (147–183)	163.2 ± 8.2 (138.8–186.6)
Median NI/BIS	43.5 ± 10.1 (23.0–68.2)	41.1 ± 5.4 (25.9–59.9)

**Table 5 sensors-23-08994-t005:** Comparison of metrics between our proposed model and other methods during three anesthesia periods in our in-house dataset with NI as the DOA.

	RMSE						MAE					
Anesthesia Period	Baseline [12]	FEDformer [45]	Crossformer [46]	Ours	Ours (Transformer)	Ours (without data preprocessing)	Baseline [12]	FEDformer [45]	Crossformer [46]	Ours	Ours (Transformer)	Ours (without data preprocessing)
All	13.30 ± 5.98	12.19 ± 3.67	11.43 ± 3.43	10.44 ± 3.23	**10.10** ± 1.61	10.64 ± 3.24	11.40 ± 6.29	9.96 ± 3.56	9.80 ± 3.56	8.76 ± 2.98	**7.52** ± 1.47	8.93 ± 3.12
Induction	15.85 ± 5.71	15.42 ± 3.87	10.95 ± 2.58	**8.01** ± 2.92	21.30 ± 4.75	8.28 ± 3.32	14.64 ± 5.97	13.47 ± 4.43	9.35 ± 2.78	6.79 ± 2.68	18.69 ± 5.70	**6.72** ± 2.78
Maintenance	12.22 ± 6.80	10.56 ± 4.24	10.66 ± 4.27	9.77 ± 3.89	**7.18** ± 2.03	10.15 ± 3.97	10.72 ± 6.97	9.15 ± 4.17	9.37 ± 4.44	8.43 ± 3.66	**5.87** ± 1.81	8.75 ± 3.86
Recovery	15.12 ± 6.72	14.60 ± 6.71	14.75 ± 6.32	13.88 ± 6.69	**10.12** ± 2.96	13.97 ± 6.13	13.80 ± 6.76	13.54 ± 6.89	13.72 ± 6.42	12.80 ± 6.86	**8.44** ± 2.72	12.93 ± 6.31
	**MDPE (%)**						**MDAPE (%)**					
Anesthesia Period	Baseline [12]	FEDformer [45]	Crossformer [46]	Ours	Ours (Transformer)	Ours (without data preprocessing)	Baseline [12]	FEDformer [45]	Crossformer [46]	Ours	Ours (Transformer)	Ours (without data preprocessing)
All	−6.02 ± 22.62	2.71 ± 17.13	6.08 ± 16.85	1.34 ± 15.65	**0.64** ± 8.02	4.67 ± 16.13	23.23 ± 12.21	19.66 ± 9.17	19.39 ± 9.09	18.29 ± 7.75	**13.08** ± 3.71	18.05 ± 8.48
Induction	15.16 ± 11.41	−14.31 ± 11.23	−6.13 ± 9.11	**−1.78** ± 7.99	−27.21 ± 13.28	−2.11 ± 8.34	17.78 ± 8.18	17.18 ± 7.79	11.33 ± 4.98	**7.74** ± 3.87	28.00 ± 12.30	8.04 ± 4.47
Maintenance	−8.77 ± 24.83	5.43 ± 18.50	8.87 ± 18.16	**1.53** ± 18.01	3.64 ± 9.08	5.63 ± 18.11	25.00 ± 13.98	19.86 ± 10.22	19.78 ± 10.80	19.36 ± 9.46	**12.10** ± 4.93	19.17 ± 10.10
Recovery	−6.04 ± 32.20	−9.73 ± 29.35	−11.95 ± 28.23	−4.07 ± 29.83	−3.98 ± 9.76	**−2.59** ± 27.44	25.49 ± 14.08	27.27 ± 17.19	28.33 ± 15.28	25.16 ± 17.46	**14.29** ± 5.97	23.46 ± 15.48

**Table 6 sensors-23-08994-t006:** Comparison of the statistic test between our proposed model and the compared methods.

	VitalDB		In-House	
Statics Test	Pairt-*t*	F	Pair-*t*	F
Ours&Baseline	0.014	0.027	0.013	0.004
Ours&Fedformer	0.007	0.036	0.003	0.044
Ours&Crossformer	0.023	0.163	0.003	0.003

**Table 7 sensors-23-08994-t007:** Comparison of the model complexity between our proposed model and the compared methods.

	Paramters (M)	FLOPS (G)
Baseline [12]	**0.1**	**0.0004**
FEDformer [45]	16.5	139
Crossformer [46]	11.4	80
Ours	9.4	0.1
Ours (Transformer)	6.2	51

**Table 8 sensors-23-08994-t008:** Comparison of the evaluation metrics between our TDM method and the boosting-based TDM method in three anesthesia periods.

	MDPE (%)		MDAPE (%)		RMSE		MAE	
Anesthesia Period	Boosting-based TDM	Ours	Boosting-based TDM	Ours	Boosting-based TDM	Ours	Boosting-based TDM	Ours
All	5.22 ± 13.25	**1.93** ± 13.36	14.91 ± 6.12	**14.53** ± 6.18	9.93 ± 2.49	**9.71** ± 2.29	7.95 ± 2.38	**7.51** ± 2.29
Induction	10.75 ± 17.54	**5.45** ± 17.83	19.01 ± 10.56	**18.64** ± 8.73	14.24 ± 5.93	**13.09** ± 4.60	12.17 ± 5.65	**11.31** ± 4.41
Maintenance	5.06 ± 13.65	**1.87** ± 13.70	14.63 ± 6.62	**14.06** ± 6.73	8.69 ± 2.44	**8.27** ± 2.42	7.19 ± 2.50	**6.75** ± 2.44
Recovery	**0.84** ± 20.88	−2.40 ± 21.02	**18.64** ± 11.74	19.44 ± 10.86	**12.75** ± 6.28	12.80 ± 5.28	**11.07** ± 5.93	11.14 ± 5.15

## Data Availability

The main data we use comes from VitalDB, available from https:vitaldb.net.

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
