# Peer review of "A Deep Learning Framework for Anesthesia Depth Prediction from Drug Infusion History"

_sensors, 2023, doi:10.3390/s23218994_

Round 1

Reviewer 1 Report

Comments and Suggestions for Authors

Here are some suggestions:

1. Abstract: Some important results need to be given

2. The authors use RNNS as the primary model. In fact, attention models also have a powerful ability to model serial data, and authors should conduct comparative analysis. For example:

[1] DSformer: A Double Sampling Transformer for Multivariate Time Series Long-term Prediction

[2] A Transformer-based Prediction Method for Depth of Anesthesia During Target-controlled Infusion of Propofol and Remifentanil

3. Evaluation indicators need to add references

4. The computational complexity of the model needs to be given

Reviewer 2 Report

Comments and Suggestions for Authors

The paper describes a deep learning framework to improve the accuracy of anesthesia depth prediction models. Although, this work is not novel, there are some interesting ideas that can further the predictive capabilities of deep learning models.

Comments

1) How do you address the inherent variability in deep learning models, as those are considered to be a black-box for a reason?

2) As a continuation from 1). The paper does not mention any robustness analysis performed on the model. 

3) The increase in accuracy, though, looks better numerically, does not seem to be a significant jump in terms of value. Why is that? How can this be improved?

Comments on the Quality of English Language

I did not see any problem in the quality of English.

Reviewer 3 Report

Comments and Suggestions for Authors

1. The manuscript focus on a study relating to deep learning based anesthesia related studies.

2. Organization of the paper, technical soundness, results and illustration is sufficiently fine.

3. Good number of references cited in the paper validates the authors proposal.

4. Architecture model presented is fine and the model is significant.

5. Results presented is accepted , but the formatting to be taken care off (E.g Table 1 & Table 6) font size seems to be unven.

6. The final section heading (References) is missing. 

7. Few interesting papers related to the context/domain could be cited in the paper:

Raju, D.N., Shanmugasundaram, H. & Sasikumar, . Fuzzy segmentation and black widow–based optimal SVM for skin disease classification. Med Biol Eng Comput 59, 2019–2035 (2021). https://doi.org/10.1007/s11517-021-02415-w
